# Seasonal Chemical Evaluation of *Miconia chamissois* Naudin from Brazilian Savanna

**DOI:** 10.3390/molecules27031120

**Published:** 2022-02-08

**Authors:** Juliana de Freitas Ferreira, Manuel Humberto Mera López, João Victor Dutra Gomes, Diegue H. Nascimento Martins, Christopher William Fagg, Pérola Oliveira Magalhães, Noel William Davies, Dâmaris Silveira, Yris Maria Fonseca-Bazzo

**Affiliations:** 1Department of Pharmacy, Health Sciences School, University of Brasília, Brasilia 70910-900, Brazil; julianafferreiraunb@gmail.com (J.d.F.F.); manuel.humberto@aluno.unb.br (M.H.M.L.); dutra.joaovictor@gmail.com (J.V.D.G.); diegue.hen@gmail.com (D.H.N.M.); perolamagalhaes@unb.br (P.O.M.); damaris@unb.br (D.S.); 2Department of Botany, Institute of Biological Science, Ceilândia Campus, School of Pharmacy, University of Brasília, Brasilia 70910-900, Brazil; acaciafagg@gmail.com; 3Central Science Laboratory, University of Tasmania, Hobart, TAS 7005, Australia; noel.davies@utas.edu.au

**Keywords:** *Miconia chamissois* Naudin, seasonality, standardized extract, Cerrado

## Abstract

*Miconia chamissois* Naudin is a species from the Cerrado, which is being increasingly researched for its therapeutic potential. The aim of this study was to obtain a standardized extract and to evaluate seasonal chemical variations. Seven batches of aqueous extracts from leaves were produced for the standardization. These extracts were evaluated for total solids, polyphenol (TPC) and flavonoid content (TFC), vitexin derivative content, antioxidant activity; thin-layer chromatography (TLC), and high-performance liquid chromatography (HPLC) profiles were generated. For the seasonal study, leaves were collected from five different periods (May 2017 to August 2018). The results were correlated with meteorological data (global radiation, temperature, and rainfall index). Using chromatographic and spectroscopic techniques, apigenin C-glycosides (vitexin/isovitexin) and derivatives, luteolin C-glycosides (orientin/isoorientin) and derivatives, a quercetin glycoside, miconioside B, matteucinol-7-*O*-β-apiofuranosyl (1 → 6) -β-glucopyranoside, and farrerol were identified. Quality parameters, including chemical marker quantification by HPLC, and biological activity, are described. In the extract standardization process, all the evaluated parameters showed low variability. The seasonality study revealed no significant correlations (*p* < 0.05) between TPC or TFC content and meteorological data. These results showed that it is possible to obtain extracts from *M. chamissois* at any time of the year without significant differences in composition.

## 1. Introduction

Cerrado, with a predominantly dry and hot climate, is recognized as the richest savanna in the world, home to 11,627 species of native plants already cataloged [1]. Cerrado comprises several phytophysiognomies and has a prevalence of some botanical families with commercial, cultural, and social importance (food, ethnobotany, religious, and others), as well as the potential for ecological restoration [2].

The richness of the Cerrado is such that the range and potential of bioactive compounds produced by Cerrado species can be considered greater than those of the Amazon Forest, representing an interesting field of investigation with medicinal plants and conservation of natural resources [3].

Melastomataceae is the sixth most abundant botanical family of angiosperms in Brazil, comprising more than 1300 species, with *Miconia*, *Leandra*, *Tibouchina*, *Microlicia,* and *Clidemia* among the most diverse genera. In all of Brazil, 267 species from the *Miconia* genus have been described and are distributed in these phytogeographic domains: Amazon, Caatinga, Cerrado, Atlantic Forest, Pampa, and Pantanal [4]. In the Cerrado, which principally covers the Midwest region of Brazil, 68 *Miconia* species have already been identified [2]. In particular, *Miconia chamissois* Naudin is a native species from Brazil found in the Caatinga, Cerrado, and Atlantic Forest [4].

Some species from Melastomataceae are used in traditional medicine as *Miconia cinnmonifolia* (DC.) Naudin and *Miconia albicans* (Sw.) Triana. *Miconia cinnmonifolia* (DC.) Naudin is used in folk medicine for the treatment of cold/fever and rheumatism. *Miconia albicans* (Sw.) Triana. is used for eupepsia. The *Tibouchina* genus is used therapeutically for pain relief. *Ossaea quinquenervia* (Mill.) Cogn. is also used in the treatment of symptomatic malarial fever by indigenous tribes of Panama [5,6].

*Miconia chamissois* Naudin is popularly known as “Folha de Bolo,” “Sabiazeira,” or “Pixirica” and has been used as food by inhabitants of Mato Grosso do Sul, Brazil [7,8]. The main phytochemical constituents of this species are anthraquinones, triterpenoids, saponins, tannins, steroids, flavonoids (rutin, isoquercitrin, and vitexin), alkaloids, and coumarins [9,10,11]. Recently, Gimenez et al. (2020) reported the presence of miconioside B, matteucinol 7-*O*-β-apiofuranosyl (1→6)-β-glucopyranoside, ursolic acid, and oleanolic acid on the chloroform partition and its sub-fraction of *M. chamissois* leaf extracts [7]. Many of these compounds are used therapeutically to treat inflammatory and neurodegenerative diseases, cancer, and other pathological processes that involve the presence of free radicals [12].

Some studies have shown the biological potential of *M. chamissois* Naudin extracts, such as antioxidant activity, *in vitro* enzyme inhibition activity against tyrosinase and alpha-amylase, antimicrobial activity [11], *in vitro* inhibition of MMP-2 and MMP-9 activities [10], and antineoplastic potential in human cervical cancer cell lines [13]. Silva et al. (2020) also demonstrated that matteucinol, isolated from *M. chamissois* exhibited selective cytotoxic against glioblastoma cell lines [14].

There has been an increase in research aimed at discovering pharmacologically active native plant species. The few studies published on the biological activity and chemical composition of *M. chamissois* have supported its therapeutic importance.

The secondary metabolism of plants is affected by temperature, ultraviolet radiation, rain index, soil nutrients, altitude, and other environmental factors [15]. *Miconia chamissois* is widely distributed throughout the Brazilian territory and therefore, is exposed to different environmental conditions.

Due to the complexity of plant species and chemical variation through seasonality, chemical elucidation and the definition of chemical markers are important for assessing the quality of the final product and ensure patient safety. Moreover, standardization of extracting extracts can reduce variability [16,17]. Thus, this study aimed to obtain a standardized extract and to evaluate the seasonal variability of *M. chamissois*.

## 2. Materials and Methods

### 2.1. Plant Materials

*Miconia chamissois* Naudin leaves were collected from Lago do Cedro, Brasília, Federal District, Brazil (coordinates 15°53′48.0″ S, 47°56′36.1″ W). A sample was deposited at the University of Brasília herbarium (UB) under the exsiccate number CW Fagg 2358. To evaluate seasonal chemical variability, leaves were collected in May 2017 (autumn), November 2017 (spring), February 2018 (summer), May 2018 (autumn), and August 2018 (winter).

Access to genetic heritage was approved by the Genetic Heritage Management Council (CGEN), registered at National System for the Management of Genetic Heritage and Associated Traditional Knowledge (SISGEN) under the number A215A9A.

### 2.2. Extraction Process

The leaves were dried at 37 °C in a circulating air oven for 30 h. The moisture content at the end of this process was 10.10% ± 0.02. The dried leaves were powdered in a knife mill to obtain a powder with 30 mesh (0.59 mm) using a sieve integrated into the mill. This powder was extracted in water by infusion at 70 °C to 50 °C at a ratio of 1:10. The aqueous extract was then lyophilized (VirTis SP Scientific Advantage Plus XL-70 Benchtop Freeze Dryer) and stored at −20 °C.

The yield of aqueous extract from *Miconia chamissois* Naudin (AEMC) was calculated as the weight percentage of the dried powdered leaves.

The total solids content was determined using an infrared moisture detector (Gehaka^®^ model IV2000, São Paulo, Brazil) from 2 mL of the sample. The analysis was performed in triplicate, and the results are expressed as the weight percentage of the dried leaf material.

### 2.3. Extraction Process Standardization

To evaluate the standardization and reproducibility of the extraction process, seven equal batches of aqueous extracts from leaves were prepared (B1–B7). Leaves collected in May 2017 at Lago do Cedro, Brasília, Federal District, Brazil (coordinates 15°53′48.0″ S, 47°56′36.1″ W) were used to prepare the seven batches. Total solids, polyphenol and flavonoid contents, TLC and HPLC profiles, vitexin derivate content, and antioxidant activity were evaluated from these batches.

### 2.4. Chemical Composition

#### 2.4.1. TLC Assay

Thin-layer chromatography was performed using silica gel (Sobernt Technologies^®^, 200 µm, 20 × 20 cm). The eluent solutions were ethyl acetate, formic acid, acetic acid, and deionized water (100:11:11:26). Chromatographic spots were visualized using both physical and chemical protocols (ultraviolet lamp at 254 nm and natural product/polyethylene glycol (NP/PEG) reagent (2% diphenylboryloxyethylamine methanolic solution—solution A and 5% polyethylene glycol ethanolic solution 4000—solution B) was used as the detection reagent [18].

#### 2.4.2. Polyphenol and Flavonoid Contents

Total polyphenol (TPC) and total flavonoid (TFC) contents in AEMCs were determined using a modified colorimetric method proposed by Kumazawa et al. (2004) [19]. The TPC assay was performed in a 96-well plate, with 50 μL of standard or sample, 50 μL of 10% calcium carbonate (Na_2_CO_3_), and 50 μL of 1N Folin reagent. One hour after the addition of Folin reagent to the reaction medium, absorbance was measured at 760 nm using a Perkin Elmer EnSpire plate reader. The results were expressed in μg equivalents of gallic acid (μg EGA/mg or percentage of total polyphenol content (%)). The TFC assay was performed in a 96-well plate by adding 100 μL of 2% aluminum chloride (AlCl_3_) solubilized in 40% ethanol and 100 μL of standard/sample. One hour after the addition of 2% aluminum chloride (AlCl_3_) to the reaction medium, absorbance was measured at 420 nm using a Perkin Elmer EnSpire plate reader. The results were expressed in μg equivalents of quercetin per mg of extract (μg QE/mg) or percentage of total flavonoid content (%). The analyses were performed in triplicate.

#### 2.4.3. HPLC-UV/DAD Assay

Chromatographic analysis by HPLC was performed using a Hitachi LaChrom Elite^®^ HPLC System (L-2130 pump, L2200 auto-sample, L-2300 column oven, and L-2455 DAD detector). Separation was carried out on a C18 column (5 µm, 250 × 4.6 mm; LiChroCART^®^150-4.6 Purospher^®^ RP18e) at 25 °C; the flow rate was set at 0.6 mL/min, the injection volume was 10 µL, and detection was performed at 354 nm. The mobile phase consisted of phosphoric acid 1% (A) (Sigma-Aldrich^®^) and acetonitrile (B) (Tedia^®^) with gradient elution performed at 0 min. 90% (A) and 10% (B); 40 min. 70% (A) and 30% (B); 50 min. 50% (A) and 50% (B). The data were analyzed using EZChrom Elite software, version 3.3.2 SP1. All solvents used were of HPLC grade (Sigma-Aldrich^®^ and Tedia^®^). The water used was obtained from a Millipore Mili-Q system. The methodology was described by Leite et al. (2014) [20]. For analysis, 25 mg of AEMC was weighed and dissolved in 5 mL of ultra-purified water/ HPLC grade methanol (6:4), which was then filtered (Hydrophyllic, 33 mm, membrane 0.45 µm). Commercial standards were used an attempt to identify the compounds present in the AEMC by comparison of retention times and ultraviolet spectra. The standards obtained were caffeic acid, chlorogenic acid, ferulic acid, kaempferol, catechin, epicatechin, gallic acid, rosmarinic acid, myricitrin, trigonelline hydrochloride, ellagic acid, isoquercitrin, hesperetin, quercetin, resveratrol, vitexin, isovitexin and myricetin acquired from Sigma-Aldrich^®^, hyperoside acquired from Hwi Analytik Gmbh^®^, and rutin from Chromadex^®^.

Vitexin equivalents (VE) were determined with a standard curve generated using vitexin in the range of 8 to 100 µg/mL. The sample was analyzed at a concentration of 5 mg/mL in a methanol and water solution (6:4). The data from the standard curve were used to estimate the vitexin equivalents in the sample using linear regression of the data. Analyses were performed for all seven batches.

#### 2.4.4. UHPLC-MS/MS Assay

UHPLC analyses were performed using a Waters Acquity H-series UPLC coupled to a Waters Acquity PDA detector in series with a Xevo triple quadrupole mass spectrometer. A Waters Acquity UPLC BEH C18 column (1.7 µm, 2.1 mm × 100 mm) was used, with mobile phases A = 0.1% formic acid and B = acetonitrile. The column was held at 35 °C with a flow rate of 0.35 mL/min, with 100% A and 0% B with a linear gradient of 40% A and 60% B at 30 min, followed by 4 min re-equilibration to the original conditions. The PDA was monitored continuously over a range of 230–500 nm. The injection volume was 15 µL. The mass spectrometer was operated in several different modes for separate injections. Initially, positive ion full scan positive ion electrospray ‘survey scans’ were acquired over the range *m*/*z* 100 to 1500 every 0.3 s, with a cone voltage of 30 V. Scanwave daughter scans at 10 V and 20 V collision energy (CE) at 2000 *m*/*z* per second were automatically acquired from the strongest ions. To unequivocally determine molecular weights, full negative ion electrospray spectra were subsequently acquired from *m*/*z* 100 to 1000 every 0.4 s using a cone voltage ramp, followed by targeted MS/MS scans with a cone voltage of 30 V and collision energy of 40 V from the relevant major [M − H]^−^ ions. The ion source temperature was 130 °C, the desolvation gas was nitrogen at 950 L/h, the desolvation temperature was 450 °C, and the capillary voltage was 2.7 KV in all cases. For this assay, a mixture of seven batches of aqueous extracts from leaves was used.

### 2.5. Antioxidant Activity

#### 2.5.1. DPPH Assay

Antioxidant activity was evaluated by reducing the DPPH (2,2-Diphenyl-1-picrylhydrazyl) radical using an adapted methodology described by Blois (1958) [21]. Ascorbic acid was used as a positive control, and a standard curve was generated over the concentration range of 10 to 1000 μg/mL. Different AEMC samples were evaluated at a concentration of 3 μg/mL. The results were also expressed as inhibition percentage (%) and were determined using the following equation:

AI (%) = 100 − (Sample Abs − Sample Blank) ∗ 100/Control Abs

AI = antioxidant inhibition (%)

Sample Abs = sample absorbance

Sample Blank = solvent

Control Abs = inhibition

#### 2.5.2. Phosphomolybdenum Method

Antioxidant activity was determined using the phosphomolybdenum method, as described by Pietro et al. (1999) [22]. Ascorbic acid was used as a positive control, and a standard curve was generated over the concentration range of 10–300 μg/mL. The AEMC was evaluated at a concentration of 125 μg/mL. The assay was performed by adding 1.0 mL of the reagent solution to 0.1 mL of the sample or the standard. The reagent solution consisted of 28 mM phosphate, 4 mM molybdate, and 0.6 M sulfuric acid. After a reaction time of 90 min in a water bath at 95 °C, the absorbance was measured at 695 nm using a Shimadzu^®^ UV-1800 (Software UVProve 2.33) spectrophotometer. The data obtained from the standard curve were used to estimate the equivalents of ascorbic acid content in the sample using linear regression.

#### 2.5.3. Lipid Peroxidation Assay

The thiobarbituric acid reactive substances assay (TBARS) was performed according to the methodology proposed by Hazzit et al. (2009), Badmus et al. (2011), and Seneviratne et al. (2016) [23,24,25] with adaptations. An emulsion of egg (10%) in 20 mM potassium phosphate buffer (pH 7.4) was prepared as a lipid source for the test. For the positive control, α-tocopherol was used at concentrations of 7.81 μg/mL to 1000 μg/mL in methanol. AEMC was evaluated over the concentration range of 3.09 μg/mL to 1000 μg/mL prepared in water:methanol solution (6:4). The percentage of lipid peroxidation inhibition was determined using the following equation:

LPI = [(Control Abs − Sample Abs)/Control Abs] ∗ 100.

Where:

LPI = lipid peroxidation inhibition (%).

Control Abs = control absorbance (solvent).

Sample Abs = sample absorbance.

### 2.6. Seasonal Study

To evaluate the seasonal chemical variability, *M. chamissois* leaves were collected in May 2017 (P1), November 2017 (P2), February 2018 (P3), May 2018 (P4), and August 2018 (P5). The development stage of the plants was observed, in which P1 was flowering/fruiting, P2 was fruiting, P3 was vegetative period, P4 was flowering/fruiting, and P5 was fruiting. The aqueous extract from these leaves was prepared by infusion following the method described above.

The meteorological data for 2017 and 2018 were provided by the Automatic Agrometeorological Station of Agroclimatology Laboratory from the University of Brasília. The data were limited to Lago do Cedro, Brasília, Federal District, Brazil, and included meteorological parameters, such as global radiation (MJm^2^ d^−1^), maximum temperature (°C), minimum temperature (°C), and rainfall index (mm) from the period of 17 January to 18 August.

Meteorological data were correlated with total solids content, TPC and TFC contents, vitexin derivate content, and antioxidant activity by DPPH assay. Pearson’s linear correlation coefficient (r) was used to determine the correlation level.

### 2.7. Statistical Analysis

Microsoft Office Excel^®^ 2016 software and GraphPad Prism^®^ Version 5.01 were used for statistical analysis. The results are expressed as the average plus standard deviation and relative deviation standard. ANOVA tests followed by Kruskal-Wallis or Dunn’s multiple comparison tests were used in different assays. Pearson’s correlation was used for the seasonal study, using linear correlations evaluated according to Callegari-Jacques considering |r| = 0–0.3 (weak), |r| = 0.3–0.6 (moderate), and |r| = 0.6–0.9 (strong) [26].

## 3. Results and Discussion

Herbal medicines have a complex chemical constitution, and their pharmacological effect are often due to synergy between these compounds. Several factors can affect the chemical composition of a plant extracts, such as growth, harvest, drying, the extraction process, and storage conditions [27]. To secure a constant composition of herbal preparations, and consequently their efficacy and safety, it is necessary to ensure their pharmaceutical quality by standardizing the process for these products [20].

In this study, a method to obtain a standardized extract of *M. chamissois* Naudin was developed, and the seasonal chemical variability was evaluated.

### 3.1. Chemical Composition and Standardization

Seven AEMC batches were prepared according to the method described above. The reproducibility of the extraction process was evaluated in relation to total solid, polyphenol, and flavonoid contents, TLC and HPLC profiles, vitexin derivate content, and biological activity by antioxidant activity.

Total solid contents found batches 1 at 7 were (2.97% ± 0.003 (B1); 2.13% ± 0.002 (B2); 2.73% ± 0.003 (B3); 2.73% ± 0.006 (B4); 2.73% ± 0.004 (B5); 2.63% ± 0.003 (B6); 2.97% ± 0.003 (B7). The average of the total solid of the seven batches was 2.70% ± 0.003, and the relative standard deviation (RSD) found was 0.11%. The results showed no significant difference (*p* < 0.05) between batches for the total solids in analysis by Kruskal-Wallis with Dunn’s test of multiple comparisons.

The extraction yields for batches 1 to 7 were 21.49%, 23.00%, 21.48%, 22.30%, 22.88%, 20.08% and 22.83%, with an average of 22.00% ± 1.06 (RSD = 4.8%).

In the TLC analysis, the same chemical profile was observed for the seven batches, with five main spots. The retention factor (R_f_) of these spots was 0.17, 0.33, 0.43, 0.54, and 0.61 (Figure 1).

The polyphenol and flavonoid analysis results are expressed as the average of the seven batches. The TP content found in AEMC was 19.65 ± 0.43 μg QE/mg (RSD = 2.19%) and the TF content was 2.49 % ± 0.15 (RSD = 5.55%).

Pearson (r) linear correlation was performed to assess the correlation between total polyphenol content and extractable solids content. A weak negative correlation (r = −0.20; *p* = 0.37) was not significant between the evaluated parameters. In general, the weak correlation indicates that the parameters evaluated are unlikely to be comparable and associate; that is, there was no correlation.

Another Pearson (r) linear correlation was carried out to correlate the total flavonoid and total solids content. The results showed a strong negative non-significant correlation (r = −0.87; *p* = 0.02), which suggested a tendency of an inversely proportional relationship between the total flavonoid content and the total solids content.

Gontijo et al. (2019) found 3.56 ± 0.17 μg equivalent of rutin/mg of extract in aqueous extract of leaves of *M. latecrenata* (DC.) Naudin prepared by infusion (1:20) [28]. In *M. albicans* (Su.) Triana fruits 510.96 ± 8.64 mg QE 100 g ^−1^ of total flavonoids was measured in July 2015, after a dry winter in the region where the fruits were collected and where the average annual temperature was 21.8 °C [29].

For other genera of Melastomataceae, *Bellucia*, in aqueous dry bark extract decoction lifting 1:10 (*p*/v) of *Bellucia dichotoma* Cogn 0.14 ± 0.030 g/100 g of flavonoids was obtained [30], and no flavonoids were detected in the aqueous extract of *Bellucia grossularioides* (L.) Triana fruits [31].

Compound detection by HPLC/DAD was performed by comparing the spectra obtained and retention times (t_R_) of seven batches of AEMC and standards. The chromatogram profiles showed 12 peaks at 354 nm (Table 1 and Figure 2). The results in the Table 1 show the mean values of retention time and peak area. The chromatogram presented in Figure 2 is representative of the analysis of one of the batches (B4).

In the identification of AEMC compounds by HPLC/DAD, the results showed similar UV spectra between peak 6 (t_R_ 24.89 min) and vitexin (0.9959) and isovitexin (0.9951) standards (Figure 2A–C); the retention times were similar. This is in agreement with the data reported by Gomes et al. (2021) [11], who found a similarity between the peak at 24.8 min of the aqueous extract of *M. chamissois* Naudin leaves and vitexin (0.9968) and isovitexin (0.9963). Peak 6 was considered to be closely related to vitexin and isovitexin.

To quantify the compound corresponding to peak 6, a linear regression of the standard curve generated using vitexin as the standard (y = 104,860x − 109,129, r = 0.99) was used to determine the vitexin equivalent (VE) content. The data analysis of seven batches showed 13.67 ± 0.57 µg VE/mg of AEMC (RSD = 4.17%).

In addition, peaks 11 and 12 showed a flavanone profile, suggesting the presence of miconioside B (t_R_ 41.36 min) and matteucinol-7-*O*-β-apiofuranosyl(1 → 6)-β-glucopyranoside (t_R_ 50.76 min), respectively (Figure 2D,E). These compounds were also identified in *M. chamissois* by Silva et al. (2020) and Gimenez et al. (2020) [7,14].

Vitexin (apigenin-8-C-glucoside) and isovitexin (apigenin-6-C-glucoside) are chemical markers for several species of *Passiflora* sp. and are present in some species of the Melastomataceae family such as *Melastoma dodecandrum* Lour. [32], *Clidemia sericea* D. Don [33], and *Dissotis rotundifolia* (Sm.) Triana [34].

For the standardization process of AEMC, the biological activity by antioxidant activity using the DPPH assay was also evaluated. The seven batches of AEMC showed 59.34% ± 7.92 (RSD = 13.35%) of inhibition of the reduction of DPPH radical at a concentration of 3 μg/mL. The results of batches B1 to B7 showed significant differences; batch B1 was significantly different from B2 and B6, B1 from B5, B4 from B5, and B5 from B6 using ANOVA test of multiple comparisons (Dunn’s) followed by the Kruskal-Wallis test.

Pearson correlation between antioxidant activity (DPPH assay) and total solids content or total polyphenol content, and flavonoid content was evaluated. The results revealed a weak positive correlation (r = 0.15; *p* = 0.42) between polyphenols and antioxidant activity and a moderate negative correlation between total flavonoids and antioxidant activity (r = −0.59; *p* = 0.14), with no significant difference (*p* < 0.05) between the chemical assays and DPPH antioxidant activity. There was a moderate positive correlation (r = 0.76; *p* = 0.06), but no significant correlation was found between antioxidant activity and total solids content.

It is possible to observe that the DPPH inhibition values for AEMC is higher than that obtained for other Melastomataceae species, considering the final concentration of 3 μg/mL with 59.34% ± 7.92 inhibition (RSD = 13.35%). DPPH inhibition by *Osbeckia aspera* var. *aspera*, *O. reticulata,* and *O. virgata* were 61.8%, 62.8%, and 60.6%, respectively, at a concentration of 100 μg/mL [35]. For *M. albicans* ethanol extract (50%) of leaves, using the same concentration of 3 μg/mL, DPPH inhibition was 36.91% ± 0.93 [36]. In addition, *Memecylon terminale* Dalz was also able to scavenge DPPH radicals with an IC_50_ value of 43 µg/mL [37].

Although the results demonstrated low variability in chemical composition, a larger variability in biological activity was observed among the seven batches tested. Nevertheless, considering the DPPH assay as a bioanalytical method, the precision should not exceed 15% (RSD) [38]. The variability in the antioxidant activity of the AEMC batches was within the allowed range.

Biological activity of several herbal medicinal products is due to the synergy between its many constituents [39]. Thus, evaluating the biological activity is an important tool for determining the quality of herbal products. Oxidative stress is present in the mechanism of many diseases, such as neurodegenerative [40], cardiovascular [41], and liver diseases [42], and in the aging process [43]. Therefore, antioxidant activity can be an important biological assay to ensure the efficacy and quality of herbal products.

#### UHPLC-MS/MS Analysis

To better characterize the chemical composition of AEMC, a UHPLC-MS/MS analysis was performed. Ten of the peaks detected matched the chromatographic profile obtained by HPLC/DAD, aiming to identify flavonoids (Figure 3). The data obtained are listed in Table 2.

The identities of compounds present in AEMC were based on the results obtained compared with data in the literature or with similarity to compounds in the Metlin MS database, and it was not possible to confirm all of them due to the complexity of the sample, and the lack of reference MS data for some relatively obscure flavonoids. In particular, the reference MS data available for the mixed C and O glycosides of apigenin and luteolin are very limited.

Peak 4 had a molecular weight (MW) of 610 with MS/MS fragments at *m*/*z* 489 [M−H-120], 429 [M−H-180], 327 [M−H- 282 (=162 + 120)], 309 [M−H-300 (=180 + 120)], and 298 [M−H-311]. A neutral loss of 282 is characteristic of 2″-O-hexosyl-6-C-hexosyl glycosides, [44,45]. In terms of the formal flavonoid glycoside fragmentation notation, the Z_1_ ion was 429, and the subsequent ^0.2^X_0_ ion was 309. These glycosides are characterized by this intense ion at the mass of the aglycone + 23. The data strongly suggested presence of an *O*-hexosyl-C-hexosyl luteolin, such as compounds like 2″-*O*-glucosyl-8-C-glucosyl luteolin (2″-*O*-glucosyl orientin) or 2″-*O*-glucosyl-6-C-glucosyl luteolin (2″-*O*-glucosyl isoorientin).

Peak 5 was also MW 610 with MS/MS fragment ions at *m*/*z* 429 [M−H-180], 357 [M−H-252 (=162 + 90)], 327 [M−H-282(=162 + 120)], 309 [(M−H-300 (=180 + 120)], and 297 [M−H-312 (=162 + 150)]. These fragment ions were consistent with the presence of two C6 sugar moieties of mixed C and O diglycosides of luteolin based on reference data and isomeric with peak 4 (Appendix A) [45,46,47,48,49].

Peaks 6 and 7 had MWs of 594 with an intense fragment ion at *m*/*z* 293 [M−H-300(=180 + 120)], while in positive ion mode, a fragment ion at *m*/*z* 433 [M+H-162] suggested the presence of mixed C and O diglycosides of apigenin (i.e., *O*-glycosides of vitexin and/or isovitexin) (Appendix A) [49,50]. These peaks together appear to correspond to peak 6 from the HPLC-UV/DAD analysis, identified as a vitexin/isovitexin derivative; we observed slightly better separation by UHPLC compared to that by HPLC.

It is noteworthy that derivatives of kaempferol, quercetin, luteolin, and apigenin have already been identified in other Melastomataceae genera, such as *Huberia* sp.*¸ Pleroma pereirae* (kaempferol and quercetin) [51], *Melastoma* sp. [52], and *Melastoma decemfidum* Roxb. (kaempferol) [53], *Osmbeckia parvifolia*, *Arnbeckia parvifolia*, *Medinilla septentrionalis* (W.W.Sm.) H.L.Li [54], *Melastoma malabathricum* L., and *M. hirta* [55].

Peak 8 with MW 432 and a prominent fragment at *m*/*z* 283 [M−H-148] was characterized as isovitexin, similar to the Metlin MS database and the literature [56,57] (Appendix A). Isovitexin has also been identified in other species of the Melastomataceae family, such as *Dissotis rotundifolia* (Sm.) Triana [34], and *Clidemia sericea*. D. Don [33].

Peak 9 had a MW of 464 with a prominent fragment in positive ion mode at *m*/*z* 303 [M+H-162], indicating loss of a hexose unit, characteristic of isoquercitrin, isoquercetin, or hyperoside [58].

Peak 12 had a MW of 594; [M − H]^−^ at *m*/*z* 593, MS/MS gave *m*/*z* 299 (neutral loss of 294), 179, 135; [M + H] ^+^ at *m*/*z* 595 with neutral loss 294 to *m*/*z* 301. A neutral loss of 294 is characteristic of the loss of both pentose (132) and hexose (162) glycoside subunits. These data and the UV spectrum indicated that it was the farrerol glycoside miconioside B, identified in *Miconia prasina* (Sw.) DC. [59]*, M. trailii* [60] and *M. chamissoi*s [7,14] (Appendix A).

Peak 13 had a MW of 608, and negative ion MS/MS gave *m*/*z* 313 ([M−H-294), indicating loss of both a pentose and hexose unit. Positive ion MS gave an intense fragment at *m*/*z* 315 [M+H-294], also indicating losses of a pentose and hexose. These data and the UV data corresponded exactly to matteucinol 7-*O*-β-apiofuranosyl (1 → 6)-β-glucopyranoside (Appendix A), previously identified as *M. chamissois* [7,14].

Peak 14 had MW 300, and MS/MS gave fragments at *m*/*z* 180, 135, and 119. This and the UV data correspond to farrerol (Appendix A), identified in *M. prasina* [59]. Yin, Jintuo et al. (2019) presented data that support to our findings [61]. Several of the very minor peaks appeared to have a farrerol aglycone component, based on the prominent *m*/*z* 301 ion in positive ion MS. Neither rutin or quercetin could be detected.

### 3.2. Antioxidant Activity

To better characterize the antioxidant activity of AEMC, two additional methods were used: the phosphomolybdenum method and the lipid peroxidation assay.

For the phosphomolybdenum assay, the standard curve of ascorbic acid yielded the regression equation y = 0.02928x + 0.0604 (r = 0.98). AEMC showed 548.8 μg EqAA/mg.

Gomes et al. (2021) found in *Miconia chamissois* Naudin equivalents of ascorbic acid content of 0.8 ± 0.01 μg/mg and equivalents of BHT content of 0.9 ± 0.05 μg/mg [11]. Murguran and Parimelazhagan (2014) found for several crude extracts from whole plant of *O. parvifolia* a phosphomolybdenum inhibition of 47.7 ± 3.2 (*n*-Hexane), 55.1 ± 7.1 (ethylacetate), 163.9 ± 10.4 (methanol) and 157.9 ± 17.5 (ethanol) mg AAE/g extract [62].

For TBARS, AEMC did not inhibit the formation of reactive species. For the dried extract (50% ethanol v/v) of *Miconia*
*albicans* leaves, an IC_50_ value of 1338.34 μg/mL [36] was observed. *Osbeckia parvifolia* Arn. ethyl acetate extract of whole plants showed 59.6% TBARS inhibition [62]. Leaves of other species, such as *Melastomastrum capitatum* A. Fern. & A. Fern. showed TBARS inhibition of 86.86% ± 3.63 and 39.93% ± 1.07 for ethanol and aqueous extracts, respectively [63].

### 3.3. Seasonality Study

Leaves of *M. chamissois* were collected in May 2017 (P1), November 2017 (P2), February 2018 (P3), May 2018 (P4) and August 2018 (P5) to evaluate the chemical seasonal variability. The results are shown in Table 3.

The meteorological data are presented in Table 4, and the correlation results are presented in Table 5.

According to the data obtained, there were no significant correlations (*p* < 0.05) between the TPC and TFC and meteorological data, with a strong negative correlation between global radiation, suggesting that the production of these metabolites is inversely proportional to the radiation index. Another strong negative correlation was observed between the antioxidant activity and rainfall index, suggesting that the antioxidant activity is inversely proportional to the rainfall index.

A moderate positive correlation between total solids and global radiation and a weak correlation between the other parameters were also observed.

Notably, the harvests were carried out in different reproductive periods, with P3 being the only harvest in which the species did not show flowering, fruiting or maturation; on 17 May, 17 November and 18 August, the highest total polyphenol content and lowest radiation were observed. Gobbo-Neto and Lopes (2007) understand the complexity of seasonality studies. In addition to meteorological and environmental parameters (water availability, radiation, temperature, soil, and altitude), the metabolic and hormonal conditions inherent in the development of the plant species must be attributed [15].

Interest in studies that assess changes in the secondary metabolism of plants has grown in recent decades and contributes to understanding the composition, adaptive capacity, and bioactive screening of species. When it comes to species from Cerrado, the interest is even greater because of the complexity and biological richness of this biome.

Studies of biological material from fauna and flora are very complex due to the object of the investigation itself, and the complexity increases when dealing with species from the Cerrado because of the diversity present in this biome. Zanatta et al. (2021) also pointed out that there may be a gap in the physiological response of plants. There will not necessarily be a positive correlation between composition and biological activity, especially with species from the Cerrado [64].

Melastomacatecae has adaptive capacity under different environmental conditions and, thus, can be present in all vegetation formations in the Cerrado [65]. Ishino et al. (2021) still emphasized the changes caused by anthropogenic issues and consequently provoked biological changes in the fauna and flora to guarantee protection. In a study of seasonality, some native plants may be stress-tolerant and may not show differences in physical and physical characteristics [66].

## 4. Conclusions

A standardized AEMC extract was obtained, and chemical marker quantification by HPLC was performed. Biological activity using the DPPH assay was proposed as an important quality parameter. In the extract’s standardization process of the extract, all the parameters evaluated showed low variability. Moreover, AEMC showed high reactivity and potential antioxidant activity via the phosphomolybdenum complex. The chemical composition confirmed the presence of polyphenolic compounds already identified in the species, contributing to the chemical elucidation of the species that have been the object of study by various research groups. Using different chromatographic techniques, luteolin glycosides, apigenin glycosides, a quercetin glycoside, miconioside B, matteucinol-7-*O*-β-apiofuranosyl (1 → 6)-β-glucopyranoside and farrerol were identified. Several of the main flavonoids were mixed C,O-diglycosides of apigenin and luteolin. The seasonal evaluation is of great value considering that there was no correlation between composition and activity versus meteorological parameters, indicating the temporal adaptability of the species. These results showed that it is possible to obtain extracts from *M. chamissois* at any time of the year without significant differences in composition.

## Figures and Tables

**Figure 1 molecules-27-01120-f001:**
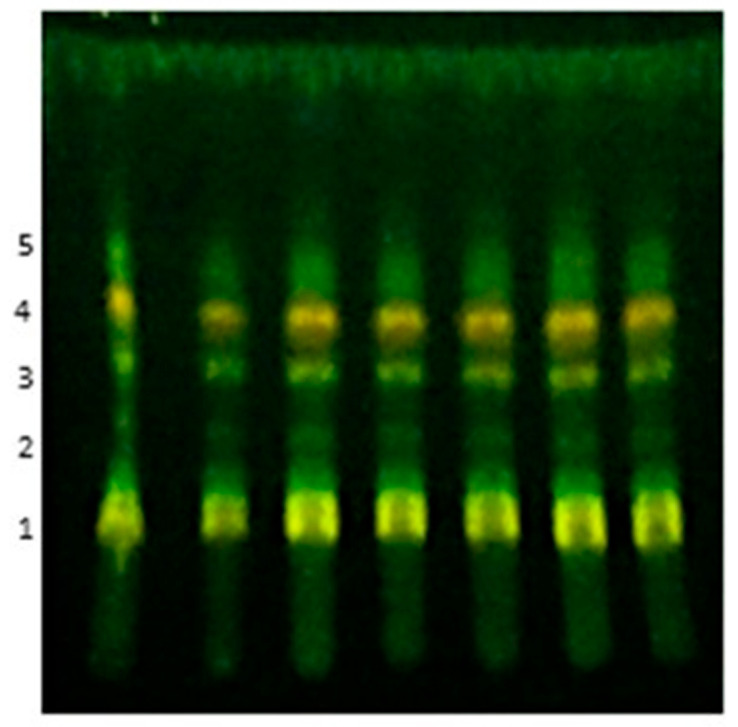
Thin-layer Chromatography from seven batches of aqueous extract of *M. chamissois* Naudin (AEMC). Each column represents one extraction batch. The leaves were collected in May 2017. The stationary phase was silica gel, and the mobile phase was a mixture of ethyl acetate, formic acid, acetic acid, and deionized water (100:11:11:26). Chromatographic spots were visualized using an ultraviolet lamp at 254 nm and the NP/PEG reagent. 1: R_f_ 0.17, 2: R_f_ 0.33, 3: R_f_ 0.43, 4: R_f_ 0.54 and 5: R_f_ 0.61.

**Figure 2 molecules-27-01120-f002:**
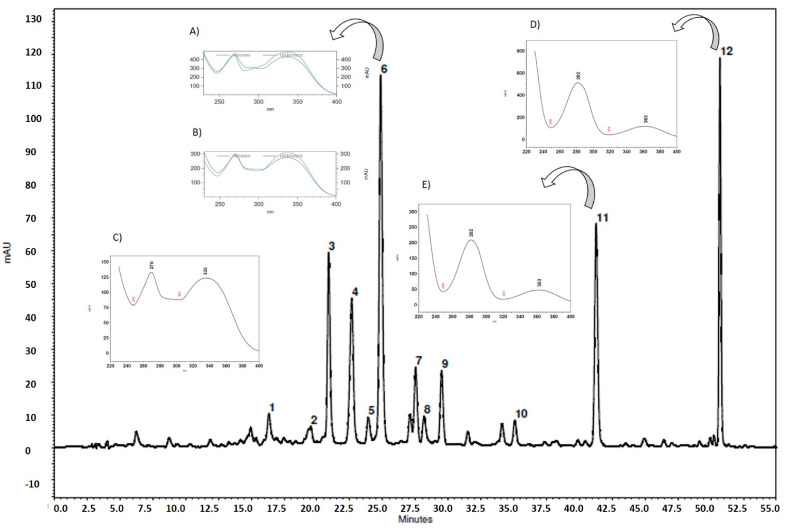
Chromatographic profile of aqueous extract *M. chamissois* Naudin leaves (B4) by HPLC/DAD at 354 nm. Detection at 354 nm, C18 column, flow rate of 0.6 mL/min, eluent: phosphoric acid 1%, and acetonitrile in gradient system. (**A**) similarity of peak 6 (t_R_ 24.89 min, λmax: 270, 336) and vitexin standard compound (similarity index: 0.9959); (**B**) similarity of peak 6 (t_R_ 24.89 min) and isovitexin standard compound (similarity index: 0.9951), (**C**) peak 6 (t_R_ 24.89 min, λmax: 270, 336); strongly suggestive a vitexin or isovitexin related compound (**D**) peak 11 (t_R_ 41.27 min; λmax: 282, 363) miconioside B; (**E**) peak 12 (t_R_ 50.76 min; λmax: 282, 363) matteucinol-7-*O*-β-apiofuranosyl (1 → 6)-β-glucopyranoside.

**Figure 3 molecules-27-01120-f003:**
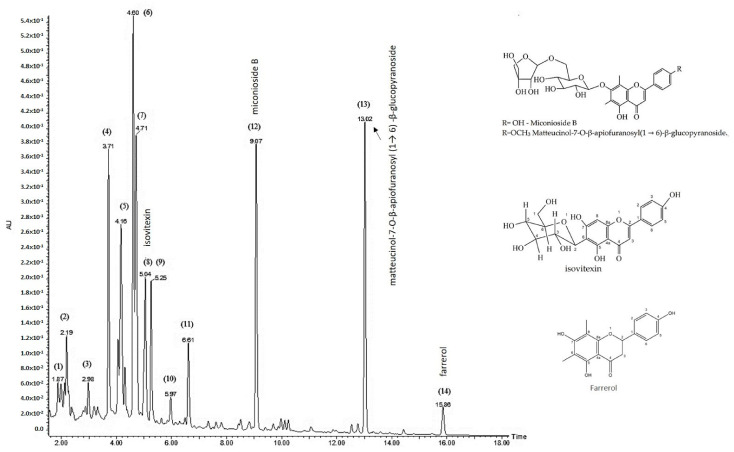
Chromatogram profile of *M. chamissois* Naudin aqueous extract from leaves (AEMC) analyzed UHPLC/UV/MS/MS at 354 nm. Detection at 354 nm, C18 column, flow rate of 0.3 mL/min, eluent: formic acid 1%, and acetonitrile in gradient system. The mass spectrometer was operated in several different modes for separate injections. The ion source temperature was 130 °C, the desolvation gas was nitrogen at 950 L/h, the desolvation temperature was 450 °C, and the capillary voltage was 2.7 KV.

**Table 1 molecules-27-01120-t001:** Peak characteristics of *M. chamissois* Naudin aqueous extract from leaves (AEMC) analyzed by HPLC/DAD at 354 nm.

Peak	Rt (min.)	Area	λ max (nm)	λ min (nm)
1	16.44 ± 0.13	605,461 ± 75,626	267; 397	393; 261
2	-	-	252; 350; 354	331; 351; 361
3	20.92 ± 0.08	3,053,209 ± 247,842	269; 349	304; 247
4	22.68 ± 0.07	3,165,255 ± 196,166	268; 256; 349	306; 246; 260
5	-	-		
6	24.89 ± 0.08	7,057,969 ± 298,667	27; 336	247; 304
7	27.56 ± 0.07	1,089,016 ± 112,057	269	250
8	-	-		
9	29.58 ± 0.09	1,169,432 ± 146,103	256; 354	319; 241
10	-	-		
11	41.36 ± 0.12	3,326,752 ± 1,134,026	28; 363	319; 249
12	50.76 ± 0.06	3,460,879 ± 1,572,479	282; 363	319; 249

Detection at 354 nm, C18 column, flow rate of 0.6 mL/min, eluent: phosphoric acid 1%, and acetonitrile in gradient system.

**Table 2 molecules-27-01120-t002:** Data from UHPLC/UV/MS/MS analysis of *M. chamissois* Naudin. aqueous leaf extract.

Peak (nº)	t_R_ (min)	UV Max (nm)		[M − H]^−^ (*m*/*z*)	MS/MS Fragments	[M + H]^+^ (*m*/*z*)	MS/MS Fragments	MW	Compound Identity or Partial Identity
1	1.87	-	-	-	-	-	-	-	-
2	2.19	-	-	-	-	-	-	-	-
3	2.98	241	-	935	-	-	-	936	-
4	3.71	269	350	609	489, 327, 309, 298	611	449, 431, 413, 383, 353	610	a mixed O,C glycoside of luteolin eg 2″-*O*-hexosyl orientin
5	4.16	255	349	609	357, 327, 309, 297	611	449, 431, 413, 353, 329	610	a mixed O,C glycoside of luteolin eg 2″-*O*-hexosyl isoorientin
6	4.6	270	338	593	293	595	433, 415, 367, 337, 313	594	a mixed O,C glycoside of apigenin eg 2″-*O*-hexosyl vitexin
7	4.71	267	338	593	293	595	433, 415, 337, 313	594	a mixed O,C glycoside of apigenin eg 2″-*O*-hexosyl isovitexin
8	5.04	269	337	431	283	433	415, 397, 367, 337, 313	432	isovitexin
9	5.25	255	354	463	301	465	303	464	isoquercitrin or hyperoside
10	5.97	276	-	599	-	-	-	600	-
11	6.61	268	341	583	447	585	449	584	?
12	9.07	281	363	593	299, 179, 135	595	301, 181, 147	594	miconioside B
13	13.02	281	362	607	313, 192	609	315, 181, 161	608	matteucinol-7-*O*-β-apiofuranosyl (1→ 6)-β-glucopyranoside
14	15.86	297	249	299	179, 135,119	301	181, 147	300	farrerol

**Table 3 molecules-27-01120-t003:** Results from seasonal study of *M. chamissois* Naudin leaves.

Collection Period	Yield (%)	Solids Content (%)	TPC (μg GA/mg)	TFC (μg QE/mL)	Antioxidant Activity (%)
P1	22.01	2.70 ± 0.003	20.73 ± 0.81	8.28 ±0.46	56.22 ± 2.44
P2	19.4	2.83 ± 0.002	19.28 ±0.49	7.70 ± 0.23	49.38 ± 4.95
P3	21.7	2.70 ± 0.002	18.58 ± 0.23	7.58 ± 0.09	47.97 ± 4.34
P4	17.94	2.70 ± 0.002	18.60 ± 0.36	7.73 ± 0.30	52.39 ± 6.09
P5	22.78	3.03 ± 0.002	19.11 ± 0.31	6.91 ± 0.21	56.60 ± 0.70

**Table 4 molecules-27-01120-t004:** Meteorological data during the collection months of *M. chamissois* Naudin.

	Global Radiation (MJm^−2^d^−1^)	Rainfall Index (mm)	Temperature Max (°C)	Temperature Min (°C)
Total solids	r = 0.363; *p* = 0.548	r = −0,092; *p* = 0.442	r = 0.430; *p* = 0.235	r = 0.198; *p* = 0.749
TPC	r = −0.775; *p* = 0.124	r = −0.190; *p* = 0.377	r = 0.230; *p* = 0.355	r = 0.209; *p* = 0.375
TFC	r = −0.753; *p* = 0.142	r = 0,10; *p* = 439	r = −0.233; *p* = 0.353	r = −0.043; *p* = 0.473
Antioxidant activity	r = 0.174; *p* = 0.780	r = −0.688; = *p* = 0.100	r = 0.012; *p* = 0,492	r = −0.257; *p* = 0.338

The data represent the monthly average.

**Table 5 molecules-27-01120-t005:** Correlation index (r) from seasonal study of *M. chamissois* Naudin leaves.

Collection Months	Global Radiation (MJm^−2^d^−1^)	Rainfall Index (mm)	Temperature Max (°C)	Temperature Min (°C)
May/2017 (P1)	332.05	45.21	22.88	17.19
Nov/2017 (P2)	337.65	241.81	23.79	18.62
Feb/2018 (P3)	363.89	160.78	23.32	20.49
May/2018 (P4)	350.38	27.69	21.02	17.42
Aug/2018 (P5)	360.6	22.86	23.38	18.64

Data represent the correlation index (r) and statistical significance by Pearson’s linear correlation test.

## Data Availability

All data included in this study are available upon request by contact with the corresponding author.

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
