# Peer review of "Seasonal Chemical Evaluation of Miconia chamissois Naudin from Brazilian Savanna"

_molecules, 2022, doi:10.3390/molecules27031120_

Round 1

Reviewer 1 Report

The article “Standardization of Extraction and Seasonal Chemical Evaluation of Miconia chamissois Naudin from Brazilian Savanna” described the composition analysis of the title plant growing in the different seasons using HPLC-DAD and UHPLC-MS/MS.  

Some revisions were suggested:

  1. The authors were suggested to check and carefully revise the typing errors. Some were highlighted in the attached file. Some remarks were also included in this file.
  2. Please provide the condition of HPLC-DAD. The authors said that the important standards such as vitexin and isovitexin are available. HPLC chromatogram of these important compounds should be included in Figure 2. This evidence would indicate the exact peaks of vitexin or isovitexin.
  3. Please provide the Figure containing the chemical structures of compounds, at least in Supporting information file.
  4. MS/MS fragmentation of detected compounds should be provided in Supporting information file.
  5. Please provide the ionization mode in MS spectra. In the method part, both negative and positive mode were provided. Please clarify.

Reviewer 2 Report

  • What is the significance of Standardization of Extraction and Seasonal Chemical Evaluation carried out in this study?
  • What are the future recommendations for this study? should be written clearly in the conclusion section.
  • Line 92: extraction was carried out in the water by infusion at 70 °C to 50°C, why this temperature range was used as 70°C is considered a relatively high temperature that may affect most of the active constituents of the plant.
  • It was better to use different solvents for extraction and different extraction methods for comparison and confirm the similarity of content in different seasons, maybe constituents will differ if the solvent was changed.

Reviewer 3 Report

In this manuscript, the authors determined in seven batches of aqueous extract from leaves of Miconia chamissois Naudin using different chromatographic and spectroscopic techniques for standardization study. The authors stated that in the extract standardization process, all the evaluated parameters showed low variability. The authors also claimed that it is possible to obtain extracts from M. chamissois at any time of the year without significant differences in composition.

Major concerns:

  1. The authors claimed that seven batches of aqueous extract from leaves of Miconia chamissois Naudin were used, however, the detailed location and seasons of each batch can’t be found in this article. The detailed location and seasons of each batch should be added.
  2. In Figure 2, the authors stated that the chromatogram presented is representative of the analysis of one of the batches. What are the location and season of the batch in Figure 2? The chromatogram of other batches should be added in supplemental materials.
  3. Which batch do Figure 3 and Table 2 present?
  4. In my opinion, this paper has nothing to do with the standardization of extraction process. Rather, only one method was used for seasonal study. The title should be changed as “Seasonal chemical evaluation of Miconia chamissois Naudin from Brazilian Savanna”. How is the method used in Section 2.2 established?
  5. The “biological activity” in Section 2.5 (line 157) should be “antioxidant activity”.

Minor concerns:

  1. The use of space characters should be checked.
  2. The “QE”, “VE”, “DPPH”, etc. in abstract shouldn’t be abbreviated.
  3. The format of reference in line 515 and 649 should be checked.

Round 2

Reviewer 1 Report

The authors have revised following the suggestion.

Reviewer 2 Report

the authors replied to all the previous comments.

the manuscript can be accepted in its current form

Reviewer 3 Report

In this manuscript, the authors determined in seven batches of aqueous extract from leaves of Miconia chamissois Naudin using different chromatographic and spectroscopic techniques for standardization study and also claimed that it is possible to obtain extracts from M. chamissois at any time of the year without significant differences in composition.

In this reply, the authors have solved most of my concerns and added more information.